# Targeting Inflammation Driven by HMGB1 in Bacterial Keratitis—A Review

**DOI:** 10.3390/pathogens10101235

**Published:** 2021-09-25

**Authors:** Linda D. Hazlett, Sharon McClellan, Mallika Somayajulu, Denise Bessert

**Affiliations:** Department of Ophthalmology, Visual and Anatomical Sciences, School of Medicine, Wayne State University, Detroit, MI 48201, USA; smcclell@med.wayne.edu (S.M.); msomayaj@med.wayne.edu (M.S.); dbessert@med.wayne.edu (D.B.)

**Keywords:** HMGB1, *Pseudomonas aeruginosa*, keratitis, silencing HMGB1, anti-HMGB1 antibody, HMGB1 box A, blockade of receptors, thrombomodulin, VIP, glycyrrhizin

## Abstract

*Pseudomonas (P.) aeruginosa* is a Gram-negative bacteria that causes human infectionsinfections. It can cause keratitis, a severe eye infection, that develops quickly and is a major cause of ulceration of the cornea and ocular complications globally. Contact lens wear is the greatest causative reason in developed countries, but in other countries, trauma and predominates. Use of non-human models of the disease are critical and may provide promising alternative argets for therapy to bolster a lack of new antibiotics and increasing antibiotic resistance. In this regard, we have shown promising data after inhibiting high mobility group box 1 (HMGB1), using small interfering RNA (siRNA). Success has also been obtained after other means to inhinit HMGB1 and include: use of HMGB1 Box A (one of three HMGB1 domains), anti-HMGB1 antibody blockage of HMGB1 and/or its receptors, Toll like receptor (TLR) 4, treatment with thrombomodulin (TM) or vasoactive intestinal peptide (VIP) and glycyrrhizin (GLY, a triterpenoid saponin) that directly binds to HMGB1. ReducingHMGB1 levels in *P. aeruginosa* keratitis appears a viable treatment alternative.

## 1. Introduction

HMGB1, originally described as a a protein that binds to DNA, functions as a structural co-factor for somatic cell transcription control [1,2]. However, it also has numerous functions extracellularly. High mobility group box 1 (HMGB1), when passively released from cells, is capable of activating host innate immunity. In that regard, it promotes dendritic cell maturation and acts as a potent proinflammatory cytokine, eliciting tissue pathogenesis and inflammation [1]. Numerous reports have suggested that it is a promising target for therapeutic intervention inanimal models of corneal infection using *P. aeruginosa*- [3,4], pneumonia in cystic fibrosis patients [1], in sepsis [5,6,7], arthritis [8], and other diseases [9]. HMGB1 is a ligand for receptor for advanced glycation end products (RAGE) and induces nuclear translocation of nuclear factor kappa-light-chain-enhancer of activated B cells (NF-κB) in macrophages (Mϕ), and neutrophils (PMN). It is found readily in the sputum of cystic fibrosis patients [1] and in the serum of septic patients, where elevated levels are consistent with poor prognosis [5]. Monocytes [10], Mϕ, [10,11], natural killer cells [12] and dendritic cells [13] secrete HMGB1 in response to engaging pathogen associated molecular patterns (PAMPS) (e.g., lipopolysaccharide, LPS), potentiating immunity.

HMGB1 structure dictates how it functions (Figure 1). It is a 215 amino acid protein with a tendency to bind LPS, and cytokines such as interleukin (IL)-1 (α and β), DNA, histones, and other molecules. Its two major receptors are Toll-like receptor (TLR) 4 and RAGE. HMGB1 acts alone or in complex with the latter molecules [14]. If HMGB1 acts on its own, its redox state, which depends on its 3 cysteines is key for a proinflammatory response. In an unstimulated cell, HMGB1 in the nucleus is fully reduced (the three cysteines express thiol groups). HMGB1 released extracellularly, in the fully reduced form, complexes with the C-X-C motif chemokine (CXC) ligand (L) 12 to enhance cell chemotaxis by binding to the CXC receptor (R) 4 [3,14]. The HMGB1 isoform with a disulfide linkage between C 23 and C 45, at the same time that C 106 remains in its reduced form as a thiol (Figure 1) allows the molecule to activate proinflammatory cytokine production. This occurs by interacting with the Toll like receptor (TLR)4 receptor, through binding to myeloid differentiation factor 2 (MD-2) [14]. Disulfide HMGB1 loses its ability to activate TLR4 in either reduced or further oxidized; reduced and disulfide isoforms are reversible. Further oxidation of HB1 generates an irreversibly converted molecule with no proinflammatory capacity [14].

HMGB1 released to the extracellular space is an attractive candidate for therapy, because of its function as a late mediator of inflammation, with levels highest levels attained at 2–3 days after infection [7]. Here, we review studies examining inhibition of HMGB1 as a therapeutic target in *P. aeruginosa* keratitis with some comparisons to other systems.

## 2. *P. aeruginosa* Keratitis: Role of Immune System in Disease

*P. aeruginosa* is a Gram-negative bacterium, often described as an opportunistic pathogen, and an important human pathogen as well. In developed countries, it remains the most common organism causing contact lens-related keratitis, one of the most rapidly developing and potentially blinding diseases of the cornea [15]. About 140 million people wear lenses worldwide, making *P. aeruginosa*-induced keratitis a global cause of visual impairment and blindness. In the USA alone, the cost of *P. aeruginosa* keratitis is approximately $175 million in direct healthcare expenditures, imposing both a clinical and an economic burden [15,16,17,18,19,20]. *P. aeruginosa* corneal infection induces inflammatory epithelial edema, stromal infiltration/destruction, ulceration and ultimately, vision loss [15].

Innate immunity exerts a major and critical component in the host pathogenic response to *P. aeruginosa* [15,21,22,23,24,25,26]. PMN and Mϕ are recruited to the infection site, engulf bacteria, and kill them by producing reactive oxygen (ROS) and nitrogen (RNS) species and facilitate their clearance [15,21,22,23,24,25]. PMN, the predominant infiltrating cell, are critical for microbial clearance [15,21,22,23,24,25]. Often, however, their persistence results in increased tissue damage and corneal perforation [21,22,23,24,25,26]. Mϕ curtail bacterial growth and regulate immune responses by controlling PMN infiltration, apoptosis and balancing pro- and anti-inflammatory cytokines and other cell responses [26,27]. This was shown by depleting mice of Mϕ by subconjunctival injection of clodronate-containing liposomes. These mice exhibited an increased influx of PMN into the cornea and more severe keratitis [27]. The adaptive immune system also is involved in this disease. It has been shown that a T helper 1 (Th1)-dominant response is associated with genetic susceptibility, severe corneal disease and perforation. In contrast, a T helper 2 (Th2)-dominant response leads to resistance, a milder course of disease and no perforation [28,29,30,31]. Recent evidence also has shown that Th17 cells infiltrate the cornea in the late stage of *P. aeruginosa* infection, sustaining inflammation, PMN influx and development of more severe disease [32,33].

## 3. *P. aeruginosa* Conventional Treatment

### Antibiotic Treatment

Treatment of *P. aeruginosa* keratitis involves intensive topical antimicrobial therapy using fluoroquinolones (e.g., moxifloxacin) or fortified Gram-negative antibiotics, including aminoglycosides (e.g., tobramycin), cephalosporins (e.g., ceftazidime), and synthetic penicillins (e.g., carbenicillin) and in severe cases, by their subconjunctival injection. Although antibiotics reduce bacterial burden, tissue damage occurs due to a poorly controlled host immune response [34,35]. Additionally, antibiotic resistant bacteria are continuously emerging and pose serious challenges for the effective management of keratitis [36]. Resistance to antimicrobials has been noted from the time the first antibiotics were discovered, and many genes that confer drug resistance upon some strains of bacteria pre-date antibiotics by millions of years [37]. However, resistance has increasingly become problematic globally due to overuse of antimicrobials which has contributed to increasing the rate of resistance development and spread. The lack of new drugs to challenge these new “supermicrobes” exacerbates the problem. Besides treatment issues, there also is an economic impact of this growing problem, as more than 2 million infections a year are caused by bacteria that are resistant to first-line antibiotics, [37] costing the US health system 20 billion dollars each year [38]. *P. aeruginosa*, an opportunistic pathogen, that does not normally infect healthy individuals, but rather those who are hospitalized or immunocompromised in some way, causes 51,000 infections/year in the USA; 13 percent [37] of these are multi-drug resistant (MDR) and increasingly difficult to treat. Other approaches, such as inhibition of HMGB1, provide novel alternatives to prevent and treat infections resulting from both resistant and non-resistant bacteria; it is both pressing and timely that we develop these alternative treatments.

## 4. Novel Approaches for Treating *P. aeruginosa* Keratitis

Figure 2 is a schematic which illustrates the approaches used for treating *P. aeruginosa* keratitis that are non-traditional. These approaches are detailed in the sections to follow.

### 4.1. Targeting Immune Cell Receptors

#### Targeting TLR4 Receptor

Others have recently shown that disulfide HMGB1 engages the TLR4 complex by binding to Myleoid (MD-2) [39]. This site differs from the LPS binding site, making it possible to selectively inhibit HMGB1/TLR4 activation without compromising LPS/TLR4 binding and its protective effects after infection [14]. This is pertinent to former studies in which we established the need for TLR4 in resistance to *P. aeruginosa* by testing a resistant inbred strain (BALB/c) of mice, whose cornea does not perforate following infection [40]. TLR4 mRNA expression was significantly upregulated in the cornea after *P. aeruginosa* infection in wild type BALB/c mice. In contrast, TLR4-deficient BALB/c mice were susceptible to infection with *P. aeruginosa* and when compared to wild type controls, exhibited increased corneal opacity, PMN infiltration, viable bacterial load, and perforated corneas. After infection, TLR4-deficient mice also showed increased mRNA expression of proinflammatory cytokines IL-1α and CXCL2 and type-1–associated cytokines interferon [(IFN)-γ and IL-18] when compared with wild type BALB/c mice. ELISA analyses showed that IL-1β, CXCL2, and IFN-γ protein levels also were significantly upregulated in the cornea of TLR4-deficient versus wild type mice. In contrast, protective levels of inducible nitric oxide synthase (iNOs) and βdefensin-2 were significantly decreased in TLR4-deficient versus wild type animals.

### 4.2. Targeting Extracellular HMGB1 Interaction

#### 4.2.1. Vasoactive Intestinal Peptide

IL-1β, CXCL2 and CXCR4, affect migration of PMNs and/or mononuclear cells into tissues. In mice that are designated susceptible to *P. aeruginosa* infection (cornea perforates) such as the C57BL/6 strain, significant upregulation of IL-1β and CXCL2 occurs post-infection (p.i.). In that regard, we have shown that vasoactive intestinal peptide (VIP), neuropeptide that reduces inflammation, boosts opposition against *P. aeruginosa* keratitis in these mice [41]. One mechanism by which this occurs is through the ability of VIP to downregulate IL-1β and CXCL2 in the cornea, resulting in a significantly reduced PMN infiltrate [41]. VIP treatment also reduced TLR-related molecules [42] consistent with their systemic reduction in a model of sepsis [43]. VIP’s use in human disease has been hampered due to delivery and other issues [44], but a side effect of VIP we discovered was to reduce HMGB1 in the infected cornea. Furthermore recombinant (r) HMGB1 blocked the beneficial effects of VIP treatment in cornea, confirming the importance of this proof of principle finding [3]. Silencing (siRNA) to lower HMGB1 levels is not amenable to clinical use, but, treatment with antagonists of HMGB1 including use of various antibodies, numerous antagonists, and other pharmacological agents, has been found to be successful in many l inflammation models of disease [44,45,46], in addition to the *P. aeruginosa* infected cornea, reducing ocular disease severity [3] (Figure 2).

#### 4.2.2. Thrombomodulin

We also tested the ability of thrombomodulin (TM) to reduce HMGB1 in the murine model of ocular bacterial infection using *P. aeruginosa* [47]. TM is an integral membrane glycoprotein and high affinity thrombin receptor found on endothelial cell membranes that functions as a natural anticoagulant. One of its domains, thrombomodulin domain (TMD) 1 has lectin-type properties, has anti-inflammatory activity and interacts with LPS associated Lewis Y antigen and with HMGB1 [48]. TM has been examined in ocular diseases that are inflammatory, such as uveitis induced by LPS [49], and similar expression is observed in the corneal epithelium and stroma in both human [50] and mouse [51] eyes. The lectin-like domain of TM reduces HMGB1, preventing deleterious effects [52,53]. In the mouse keratitis model [47], as in silencing HMGB1 (siRNA use), recombinant (r) TM, composed of TM domains 1-4, (Leu 17-Ser 517), treatment led to better clinical disease scores in treated mice after after infection and slightly (1 log) lessened the plate count in cornea. Treatment also diminished mRNA levels for proinflammatory agents (NF-κβ, TLR4, and RAGE). In addition, it led to a significant, albeit modest, upregulation in anti-inflammatory cytokines including single immunoglobulin IL-1R-related receptor (SIGIRR) [54] and suppression of tumorigenicity 2 (ST2) [55] which contribute to better disease outcome. Nonetheless, HMGB1 (mRNA or protein) was not decreased. Soluble TM also binds to HMGB1 and takes part in the thrombin-mediated proteolysis and cleavage of HMGB1 [53]. Recombinant TM is currently used in Japan as a treatment for septic patients with disseminated intravascular coagulation [56].

### 4.3. Targeting Extracellular HMGB1 Protein Directly

#### 4.3.1. Silencing HMGB1 by siRNA

HMGB1 recruits inflammatory cells through CXCL12, which forms a heterocomplex with a fully reduced HMGB1 isoform to act exclusively through CXCR4 and promotes mononuclear cell migration [57]. If monocytes and Mϕ were affected by RNA interference of HMGB1 in the keratitis model was unknown and so after silencing HMGB1 [3], we examined amounts of CXCL12 and CXCR4 (mRNA and protein); both were reduced vs. scrambled control levels. Markers for monocytes/macrophages (MOMA+) and only Mϕ (F4/80+) were used separately to label and quantitate the cells. Positive cells for each of the markers were calculated and shown as a percent of all corneal-infiltrated cells. By 5 days p.i., silencing had reduced both MOMA+ cells (monocytes and Mϕ) and F4/80+ cells (Mϕ only) significantly compared with controls that received scrambled siRNA [3]. These studies agree with past work that showed heterocomplex formation inhibition with a chemical inhibitor, plerixafor (AMD3100) [57,58], a CXCR4 receptor antagonist, curtailed monocytic cell recruitment into air pouches and injured muscles [57]. In the infected cornea, PMN infiltration was not regulated by this mechanism [3].

#### 4.3.2. Anti-HMGB1 Antibody

We used a commercial neutralizing polyclonal antibody [anti-HMGB1 antibody (IBL International, Toronto, ON, Canada] made from a synthetic peptide to amino acids (KPDAAKKGVVKAEK) of human HMGB1 and showed that prophylactic antibody treatment improved the *P. aeruginosa* infected C57BL/6 mouse cornea by decreasing the clinical disease score, the PMN infiltrate and perforation when compared with control antibody treatment [3]. These results were confirmed using resistant (no perforation after similar infection) BALB/c mice who were treated with recombinant (r) HMGB1. These mice had more severe disease after rHMGB1 injection when compared with controls [3]. Other complimentary work has shown that an anti-HMGB1 neutralizing monoclonal antibody (mAb) prevented death due to sepsis and acute liver injury and inhibited HMGB1 endocytosis. The latter, results in Mϕ pyroptosis if it binds to RAGE and is internalize by a receptor mediated endocytic mechanism [14,59,60]. Another anti-HMGB1 mAb was developed in Japan [61] and recognizes an epitope in the C terminal sequence of HMGB1. Success in therapeutic intervention in several neuroinflammatory (e.g., stroke, spinal cord injury, epilepsy) conditions has been established in animal studies.

#### 4.3.3. HMGB1 Box A Protein

Human HMGB1 is composed of three major functional domains: A box, B box, and the C-terminal acidic tail (C tail) [62,63] (Figure 1). Box B (first 20 amino acid residues of this domain) has proinflammatory cytokine-inducing capability [64]. Only Box A competitively inhibits HMGB1 binding to its receptors and attenuates Box B and its proinflammatory effect [14,45,65]. The C tail contributes to the spatial structure of both Box A and Box B and regulates HMGB1 DNA binding [5,66]. Resembling the work using anti-HMGB1 antibodies, animal studies revealed that HMGB1 Box A treatment prevented the proinflammatory cytokine effects of HMGB1 and improved both infectious and non-infectious diseases [67]. In mouse models of sepsis, Box A treatment improved disease outcome and increased survival [45]. Box A not only antagonizes HMGB1 binding, but also reduces HMGB1-induced release of proinflammatory cytokines, [68] both of which suggest strong therapeutic potential. Studies also demonstrate that even delayed treatment with Box A is capable of specific inhibition of endogenous HMGB1 and therapeutically reversed lethality of established sepsis [45].

In the *P. aeruginosa* infected cornea, HMGB1 Box A recombinant protein (89 amino acids) versus PBS prophylactic or therapeutic treatment regimens significantly reduced clinical scores, myeloperoxidase (MPO), CXCL2 activity, bacterial load, and expression of TLR4, RAGE and other cytokines. Box A blocked colocalization of HMGB1 interaction with TLR4 and RAGE on infiltrated cells in the infected corneal stroma and reduced protein levels of IL-1β, CXCL2, and IL-6 [68]. The recent identification of Box A blocking RAGE-mediated uptake of HMGB1 and complexes provides the ability to evaluate Box A activity in vitro and enhances its clinical development [14].

#### 4.3.4. Glycyrrhizin and Carbenoxolone

Glycyrrhizin (GLY) and carbenoxolone, a synthetic derivative, are able to bind HMGB1 causing reduction in a chemokine and cytokine mediated inflammatory cascade [69,70,71]. GLY is a plant glycoside obtained from licorice roots (*Glycyrrhiza glabra*) and a saponin structurally. It has many pharmacological effects [72] and is successful when used for treatment of sepsis, [73] colitis, [74], lung [75], and brain [76] injury in animal models. It was also found useful in the clinical management and treatment of chronic hepatitis patients [77]. Carbenoxolone is an anti-inflammatory drug that is authorized for use in treatment of peptic ulcers, [78]. It also is efficacious when used in animal models for lung [79] and ischemic brain injury [80]. When we tested the two agents in the keratitis model [4], GLY was better overall in decreasing HMGB1 expression (mRNA and protein) in KEI 1025 (a clinical isolate of *P. aeruginosa*)–infected corneas and decreased corneal disease better when compared with carbenoxolone. Better outcome to disease was consistent with reduced critical cytokines, IL-1β and CXCL2 (mRNA and protein), PMN influxion, bacterial plate count, and increased levels of antimicrobial (mouse beta defensin 2) protective protein. We also tested GLY after infection with American Type Culture Collection (ATCC) 19660, a cytotoxic strain of the bacteria. Treatment similarly reduced keratitis after infection with this ATCC strain. Correlating with less pathology, we found decreased CXCL2 protein, PMN infiltrate, and bacterial plate count which was effective even when treatment was begun 6 h after infection [4]. We tested whether GLY was effective on a drug resistant isolate, resistant strain 1 (RS1) of *P. aeruginosa* which proteomics showed had a significant increase in expression of outer membrane efflux pump and fimbrial proteins compared with a control strain PAO1 [81]. GLY significantly reduced RS1 biofilm formation and binding to mouse and human corneal epithelial cell cultures and in vivo, contributed to better clinical scores, reduced PMN infiltrate and bacterial load, consistent with previously observed results. We then tested GLY against a multi-drug resistant non-ocular isolate, MDR9, which was resistant to 6/12 antibiotics tested including ciprofloxacin [82]. This study showed that GLY was effective against MDR9 when used in combination with ciprofloxacin, and most significantly, remained effective when treatment was not inititated until 18 h following infection. Additionally, ciprofloxacin’s minimum inhibitory concentration (MIC) (32 μg/mL) was decreased 2-fold to 16 μg/mL when ciprofloxacin and GLY were used together. GLY affected bacterial membrane permeability and decreased viability. It also was the most effective of any of the HMGB1 inhibitors we have tested as it optimally reduced clinical score, bacterial load and PMN count assessed by Myeloperoxidase (MPO). Recent (unpublished) data show that GLY is effective at acidic (4.0) and neutral (7.2) pH in the keratitis model.

## 5. Closing Remarks

This review has provided information that inhibition of HMGB1 can be successfully used in both rodent and humans to modulate disease. Our focus on microbial keratitis supports the tenet that inhibition of HMGB1 is successful at reducing keratitis in rodent models and that one of the inhibitors, GLY also potentiates antibiotic treatment against both non, as well as multi-drug resistant isolates, even when treatment is begun almost a day after infection. Testing of GLY or other inhibitors in patients is the logical next step to determine if all the observations reported in rodents will translate to human care.

## Figures and Tables

**Figure 1 pathogens-10-01235-f001:**
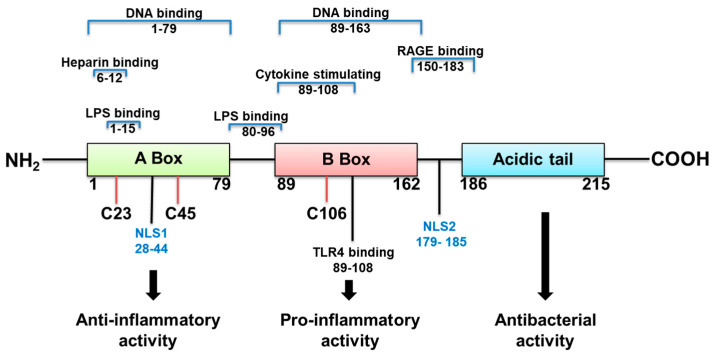
HMGB1 structure, binding sites and function.

**Figure 2 pathogens-10-01235-f002:**
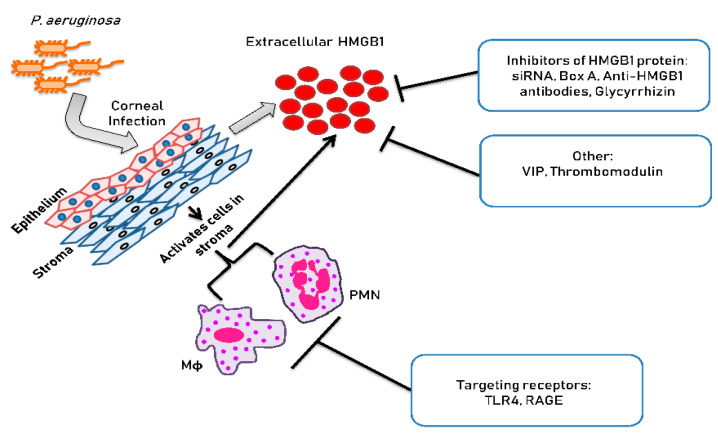
Approaches to inhibit HMGB1 in cornea.

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
