# Peer review of "Targeting Inflammation Driven by HMGB1 in Bacterial Keratitis—A Review"

_pathogens, 2021, doi:10.3390/pathogens10101235_

Round 1
Reviewer 1 Report
Pahtogens-1340530-Review
In this review, the authors describe an alternative approach for treating bacterial infections; focusing on targeted inhibition of one promising target, high mobility group box 1 (HMGB1), a member of the alarmin 17, and prove this approach for treatment of keratitis, a severe infection of the eye; caused by P. aeruginosa.
The review is very focus on the topic, but at the same time very comprehensive in detailing new treatment approaches for targeted inhibition of HMGB1. However, a few major comments have to be address to enhance its value and to a boarder audience of readers.
- The organization of the review is a bit “clumsy” and mix topics; also caused by the unsuccessful names of the sub-headings; thus rendering the manuscript very hard to follow.
* Sub-heading 2 is misleading in its name; should be “P. aeruginosa keratitis and involvement of the immune system in the disease”. The next subheading should be:
3. P. aeruginosa keratitis and conventional treatment;
3.1 Antibiotic treatment
Organizing the following subheadings according to the authors’ own scheme in Figure 1 and reversing the order of its appearance:
4. Novel approaches for treating P. aeruginosa keratitis
4.1 Targeting immune cells’ receptors
4.1.1 Targeting TLR4 receptor (RAGE?)
4.2 Targeting extracellular HMGB1’ interactions
4.2.1 VIP
4.2.2 Thrombomodulin
4.3 Targeting directly the extracellular HMGB1 protein
4.3.1 Silencing HMGB1 by siRNA
4.3.2 Anti HMGB1
4.3.3 HMGB1 Box A protein
4.3.4 Glycyrrhizin and Carbenoxolone
In this outline, the review of treatments start from the more general treatments and ending in the most specific ones; Glycyrrhizin and Carbenoxolone being the ones the authors themselves choose to highlight in their conclusions.
* As HMGB1 is the major topic of this review, it is strongly suggested that a scheme describing its intra- and extra-cellular activity, including its interactions with the immune system components, should be added (detailing also its redox state in each case).
* The manuscript is full of abbreviations that are not fully described; see for example RAGE, CXCR4, CXCL12, MD-2 and many others. Authors should carefully review the whole manuscript and provide full names for each abbreviation mentioned for the first time.
* In the original 3.1 subheading, please indicate that silencing of HMGB1 was performed by siRNA. Similarly, in the original 3.3 sub-heading, please indicate directly what is the real treatment with HMGB1 Box A? Is it indeed treatment with a protein? peptide? What is the size of the box A domain?
Author Response
8/27/2021
[Pathogens] Manuscript ID: pathogens-1340530 - Major Revisions
Our response to review is as follows we have made all changes requested:
Reviewer 1. Organization of reviews suggestions. This was not shown in track changes and the bibliography changes were not either as they are massive and would only confuse. The other changes are in track changes for easy view.
- We have reorganized the headings and subheadings with advice from this reviewer as recommended and to the best of our ability. (not shown in track changes)
- We have included a scheme as requested showing the interactions of HMGB1, intra and extracellular activity and immune interaction-in general; redox state as well. This is a new figure (new Figure 1).
- We checked all abbreviations-and provide full name first use (track changes shown)
- We indicated that siRNA was used; 3.3 we will indicate the treatment with Box A as specific (recombinant protein) and the size of the box A domain (89 amino acids).
Reviewer 2 Report
This is a concise, informative and well-written review that summarises attempts to target HMCB1 in bacterial keratitis. I think that it would be enhanced by including a simple diagram showing the domain structure of HMGB1 and the position of the cysteines and the importance of these on its redox state. Also, a very minor point, reference [3] should be given at the end of the sentence (line 11, section 3) .."we discovered was to reduce HMGB1 [3]".
Author Response
Reviewer 2: Thanks very much for the positive comments that it is informative and a well-written review-we appreciate it.
- We agree, it is enhanced by a diagram showing the domain structure of HMGB1, cysteines position and importance of redox state. We have added that as a Figure (new Figure 1).
Minor point: reference 3 moved to end of sentence as requested.
Round 2
Reviewer 1 Report
The authors addressed comments and correct the manuscript accordingly.